# Aerobiological monitoring in a desert type ecosystem: Two sampling stations of two cities (2017–2020) in Qatar

**Maryam Ali Al-Nesf**[1]*, **Dorra Gharbi**[1,2], **Hassan M. Mobayed**[1], **Ramzy Mohammed Ali**[1], **Amjad Tuffaha**[3], **Blessing Reena Dason**[1], **Mehdi Adeli**[3], **Hisham A. Sattar**[4], **Maria del Mar Trigo**[2]

**1** Allergy and Immunology Division, Hamad Medical Corporation, Doha, Qatar, **2** Department of Botany and Plant Physiology, University of Malaga, Campus de Teatinos, Malaga, Spain, **3** Sidra Medicine, Ar-Rayyan, Qatar, **4** Pulmonary Division, Hamad Medical Corporation, Doha, Qatar

\* mariam_alnisf@hotmail.com

## Abstract

### Background

The increasing number of aerobiological stations empower comparative studies to determine the relationship between pollen concentrations in different localities and the appropriate distance, which should be established between sampling stations. In Qatar, this is basically the first aerobiological study for a continuous monitoring interval.

### Objectives

The study aimed to assess the abundance and seasonality of the most prevalent pollen types, plus identify potential differences between two sites within the country.

### Methods

Airborne pollen data were collected during 2017–2020 by using Hirst-type volumetric samplers in Doha capital city and Al Khor city in Qatar, placed 50 km apart.

### Results

Higher total pollen indexes were recorded in the Al Khor station (2931 pollen * day/m³) compared to the Doha station (1618 pollen * day/m³). Comparing the pollen spectrum between the sampling stations revealed that ten pollen types were found in common. Amaranthaceae and Poaceae airborne pollen constituted 73.5% and 70.9% of the total amount of pollen detected at the samplers of Al Khor station and Doha station. In both sampling sites, a very pronounced seasonality was shown; August–October appeared as the period with the most intense incidence of atmospheric herbaceous pollen, with 71% and 51% of the annual total counts in Al Khor and Doha stations, respectively. August (Al Khor, 21%; Doha, 9%), September (Al Khor, 33%; Doha, 26%), October (Al Khor, 17%; Doha, 16%) were the months in which the herbs pollen concentrations were highest.

**Data Availability Statement:** All relevant data are within the paper. All datasets used or analyzed in this study are present in the paper in their raw form.

**Funding:** This paper concluded out of 3 years grant project "Aerobiological studies in Qatar and Sharjah: Toward the establishment of a network for pollen analysis and allergenicity" (NPRP 9-241-3-043). We declare that this project received supported grant funding from the Qatar National Research Fund (QNRF). MAA received the awarded grant. URL: https://www.qnrf.org/en-us/Funding/Research-Programs/National-Priorities-Research-Program-NPRP. The funders had no role in study design, data collection and analysis, decision to publish, or preparation of the manuscript.

**Competing interests:** We like to disclose that the research reported in this manuscript received supported grant funding from the Qatar National Research Fund (QNRF)-Qatar (NPRP 9-241-3-043). However, the funders had no role in study design, data collection, and/ or analysis, decision to publish, or preparation of the manuscript. Also, we like to confirm that this does not alter our adherence to all PLOS ONE policies on sharing data and materials. There are no other commercial associations that might pose a conflict of interest and no known conflicts of interest associated with this publication.

**Abbreviations:** APIn, The Annual Pollen Integral; ESA, The European Society of Aerobiology; SPSS, Statistical Package for Social Sciences software.

Significant statistical differences between the two stations were observed in specific pollen types with local distribution in each trap's vicinity.

## Conclusions

Comparison of data obtained by the two samplers running at a distance of 50 Km indicated that potential inter-site differences could be attributed to the vegetation surrounding the city having a decisive influence on data collected.

## Introduction

Knowledge of the atmospheric pollen concentration in a particular station linked to the micrometeorological indicators can guide in establishing a chronological correlation between the concentration of pollen grains in the air and the symptoms of the different allergic disorders such as allergic rhinitis and asthma and further lead to correct etiological diagnosis and model appropriate management approach and treatment plan [1, 2].

The pollination season and duration vary between countries depending on the climate and the vegetation [3, 4]. It might differ between cities [5] or areas of a single city [6] due to distinct bioclimatic and biogeographical conditions in these areas. Studies focusing specifically on different areas of a given city noted that vegetation surrounding the city could have a decisive influence on the collected data [7].

Airborne pollen counts are commonly determined using a volumetric suction sampler, such as the Hirst-type spore trap, functioning based on the impact principle [8]. Due to their volumetric nature, these samplers can detect pollen from a larger area; therefore, a single sampler is commonly used to provide airborne pollen counts for the whole city, leading to establishing the pollen calendar. The pollen calendar is a graphic representation that summarises all the aerobiological information of a particular location in a single picture [9] and can provide data regarding different levels of airborne pollen concentrations and may help in prognosis and preventing the incidence of allergic reactions symptoms [10, 11]. In addition, regional pollen calendars are an effective tool used in clinical allergy facilities to establish a management approach and choose more personalised treatment [12]. Identifying the occurrences of airborne pollen within the city and across different cities should be an essential part when considering establishing a pollen calendar in an area. The usefulness of such an approach is signified by determining the spatial variation of atmospheric pollen levels and delimiting living conditions in specific areas in the city or even different cities based on the risk of sensitization to the known pollen allergies. Moreover, identification of patients with sensitization to pollens in the different allergy clinics located in different cities might be related to differences in the macro-environmental (i.e., climate) and microenvironmental (i.e., city) that are specific to that area [13, 14].

In the Middle East and North Africa region, aeropalynological research and pollen calendars in urbanised areas are scarce or fragmentary. In Qatar, pollen calendars have never been assessed. Among the countries in this region, Kuwait [15] was the origin of the first pollen calendar providing baseline data on the prevailing airborne pollen types in the country, followed by Jordan [16, 17], Saudi Arabia [18] and Egypt [19].

A pilot study was launched in May 2017 and highlighted establishing a national aerobiological network to investigate the aerobiological airborne and the prevalence of common pollen sensitisation in the state of Qatar. Two pollen-sampling stations with continuous flow

volumetric methods for pollen monitoring following the International Association for Aerobiology recommendation were used. Data about the airborne pollen content, the seasonal distribution over the year, and the effect of short-term exposure of pollen on allergic respiratory symptoms in the atopic population were collected. The survey of the different pollen in the atmosphere of Doha and Al Khor cities for the two full years delineated the various botanical families likely to contribute to allergic diseases in the peninsula of Qatar and emphasised the importance of having such studies even in the desert climates [20].

The current study aimed to establish the first pollen calendar in two main cities in Qatar and assess the type and concentrations of airborne pollen content between two localities (Doha and Al Khor) over three years of continuous aerobiological monitoring. More specifically, we aimed to investigate the intra-annual and seasonal behaviour of the main allergenic pollen types obtained from the two sampling stations of Doha and Al Khor cities.

## Materials and methods

### Study area

The city of Al Khor, with almost 202,031 inhabitants approximately, is one of the largest cities in Qatar state. It is on the northeast coast of Qatar (25˚41′24″N, 51˚30′36″E), around 50 kilometres from the capital, Doha (25˚17′12″N, 51˚32′0″E), with 641,000 habitants [21]. Geologically, the peninsula of Qatar is characterised by aeolian sandy soils, which are characterised by a deep profile with calcareous coarse sand to loamy coarse sand and admixture of the desert and marine sand [22].

The natural vegetation around both cities is mainly shrubs, perennials, and ephemerals. The vegetation cover is very low (average 1–25%). The plant communities are characterised by the dominance of halophytic and xerophytic coastal communities such as *Salsola imbricate*, *Suaeda aegyptiaca*, *Suaeda vermiculata*, *Halopeplis perfoliate*, *Zygophyllum* spp., *Cyperus conglomeratus* with *Fagonia indica* and *Arnebia hispidissima*, among others as the main associates [23]. Climatologically, Qatar has a sub-tropical desert-type ecosystem and arid climate [24]. The mean minimum monthly temperature of the coldest month of the year, January, is 13.4˚C, while the mean maximum of the hottest month, July, is 40.7˚C. The relative humidity is high throughout the year, with an annual mean of 65%. Rainfall is very erratic and irregular in time and space, influencing the type and amount of vegetation in the desert [22].

### Pollen sampling

The daily mean concentrations were monitored from May 2017 to May 2020 using a Hirst volumetric trap (VPPS 2000 Lanzoni, Bologna, Italy). The pollen traps were placed on the rooftop of Hamad Medical City building (Doha) and Al Khor Hospital (Al Khor), 1 m above the roof and 20 m above the ground level, ensuring no physical barriers and free air circulation. The air sampling was carried out from May 7, 2017, to May 7, 2020, in Doha city, and from May 8, 2017, to May 8, 2020, in Al Khor city. The sampling stations are 50 Km apart (Fig 1). The samplers operated continuously, aspirating a constant flow of 10 L per min and impacting the atmospheric particles on a Melinex® tape impregnated with silicone fluid. After exposure, the tape is cut into 24-h fragments (48 mm) and mounted on slides using glycerin jelly. The drum and tape were removed weekly and replaced with another drum every Sunday (at 10 am; Doha station) and Monday (at 7 am; Al Khor station).

The daily number of pollen grains per cubic meter (expressed as pollen grain/m$^3$) was determined and counted along four horizontal transects with Optika light microscopy at x400 magnification. The method was standardised and used for pollen counting and data interpretation based on the Spanish Aerobiology Network [25] and the minimum requirements from

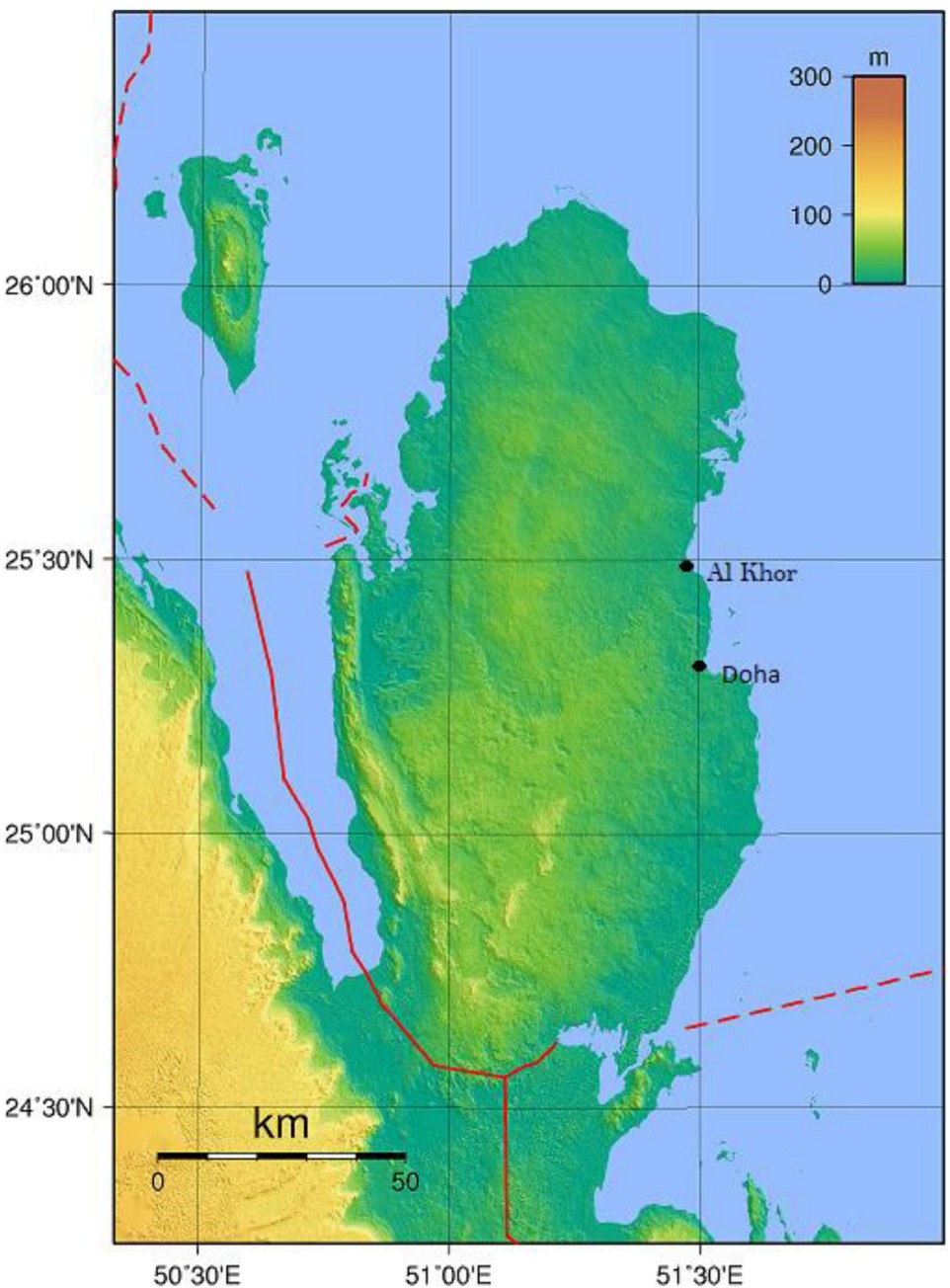

**Fig 1. A topographic map of Qatar showing the main sampling locations.**

the European Society of Aerobiology (ESA) [26]. Pollen grains that could not be identified were defined as undetermined types. The Annual Pollen Integral (APIn expressed as pollen * day/m³), representing the sum of the daily mean pollen concentration over the given period, was calculated following aerobiological studies' recommended terminology [27]. Pollen types that comprised more than 1% of the annual total pollen concentration were considered dominant. The meteorological data for the years of the study were provided (Fig 2). These were the mean of the minimum and maximum temperatures (˚C) and mean of total rainfall (mm) in the two cities, Doha and Al Khor.

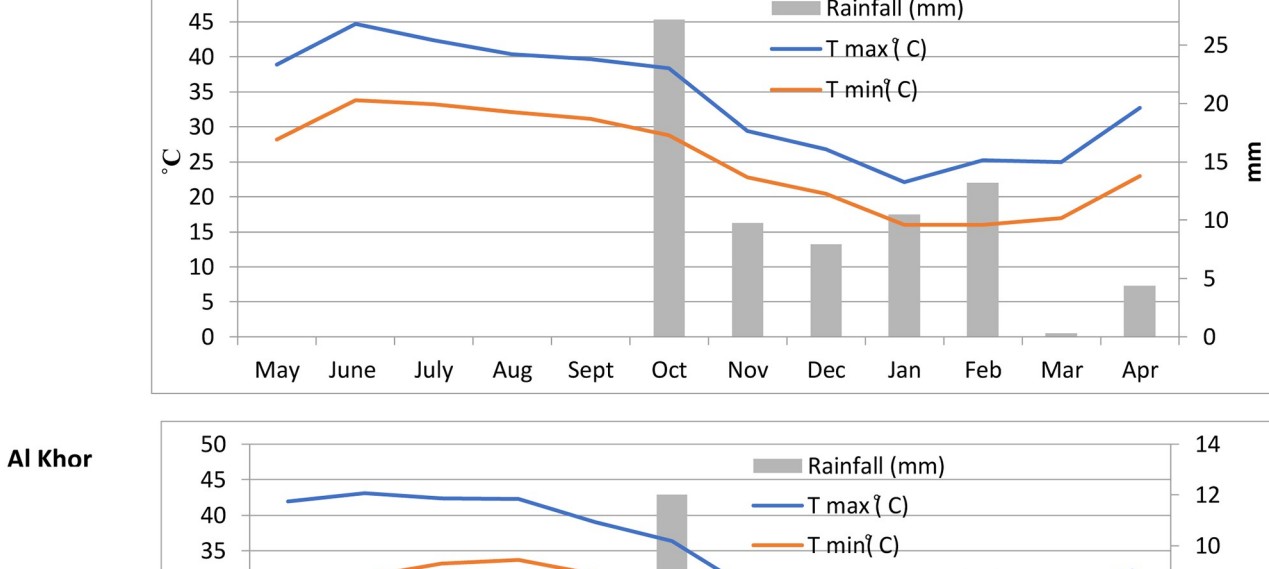

**Fig 2. Mean of meteorological parameters (minimum temperature (˚C), maximum temperature (˚C), and rainfall (mm)), taken at the sampling sites (2017–2020).**

### Qatar pollen calendars

The pollen calendar was constructed based on the mean of total weekly counts of pollen/m$^3$ in 2017–2020 [28, 29]. In the calendar, pollen grains per m$^3$ were graded as low (1–9), moderate (10–49), and high ($\geq$50).

### Statistical analysis

The daily mean pollen concentrations of a dominant taxon found in common between the sampling sites and the annual pollen integrals for the entire period (3 years) were calculated. Data normality was evaluated by the Shapiro–Wilk test (W). Since the data did not adjust to normal distributions, non-parametric statistic tests were applied. The spearman rank correlation coefficient was used to establish the relationship between the two samples' daily pollen counts. The Wilcoxon test was used to search whether the differences observed were significant. Statistical Package for Social Sciences software (SPSS Chicago IL, USA) for Windows, Version 21.0 was used for the analysis, and a P$\leq$.05 was considered statistically significant.

### Ethics approval and consent to participate

The study was approved by the Ethical Committee of the Hamad Medical Corporation, Doha, Qatar (MRC#16150/16). All clinical investigations were conducted according to the principles

expressed in the 1964 Helsinki declaration and its recent amendments. The current study was not involving patients or animals. However, the QNRF project associated with this study involved patients' data and written informed consent was obtained from all the participants in the project.

## Results

### Differences between the two stations

Quantitatively, significant differences were observed between the individual years within the individual pollen station and between the two stations. During the three monitoring years (2017–2020), the APIn of the total pollen grains in Al Khor was almost two times higher than the APIn of Doha (2931 and 1618 pollen * day/m$^3$, respectively). The difference was also seen among the different years, with the highest value was attained in 2019–2020 (Al Khor, 1603 pollen * day/m$^3$; Doha, 1011 pollen * day/m$^3$) and the lowest in 2018–2019 (Al Khor, 364 pollen * day/m$^3$; Doha, 275 pollen * day/m$^3$). Overall, there was high variability in APIn and the maximum total pollen concentration for each specific type (Table 1). Considerable differences in daily mean pollen concentrations were observed between the three studied years and the aerobiological stations considering year-to-year seasonal behaviour (Fig 3).

The pollen spectrum includes 25 different pollen types identified in the Al Khor locality (14 herbaceous and 11 trees), while 24 pollen types (14 herbaceous and 10 trees) were recorded in the Doha area. In both areas, the highest percentages were notes for herbaceous plants (Al Khor, 87%; Doha, 86%), followed by trees (Al Khor, 13%; Doha, 14%). A total of 12 abundant pollen types were detected in Doha, and 11 pollen types with significant influence (>1% total annual) were found in Al Khor.

Comparing the pollen spectrum between the sampling stations revealed ten pollen types were found in common, in order of abundance: Amaranthaceae, Poaceae, *Plantago*, *Artemisia*, Cyperaceae, and Brassicaceae among the herbaceous group and *Prosopis*, *Conocarpus*, Araceceae among the trees group. *Urtica* and Cupressaceae pollen represented the dominant pollen only at the Doha station, and Fabaceae pollen was much more frequent at the Al Khor station. Of all the pollen types, pollen grains of the Amaranthaceae and Poaceae families constituted 73.5% and 70.9% of the total amount of pollen detected at the samplers of Al Khor station and Doha station, respectively (Table 1).

In both sampling sites, a very pronounced seasonality was shown; August–October appeared as the period with the most intense incidence of atmospheric herbaceous pollen, with 71% and 51% of the annual total counts in Al Khor station and Doha station, respectively. August (Al Khor, 21%; Doha, 9%), September (Al Khor, 33%; Doha, 26%), October (Al Khor, 17%; Doha, 16%) were the months in which the herbs pollen concentrations were highest. On the contrary, the lowest pollen concentrations were always registered in May, June, and July, increasing again in early November. In March, pollen concentrations increased gradually, showing a peak of 15% of annual pollen counts in the Doha station, while only 7% of annual pollen counts were observed in Al Khor (Fig 4).

Furthermore, an analysis of the monthly dynamics of trees pollen types in both sampling sites depicted different pollen curves shapes for a particular period. The highest pollen count in Al Khor was recorded during two periods. The first was three continuous months (from September to November) in Al Khor, while in Doha, the peaks took place in September and November. The second period of the high concentrations took place in March in both stations (Fig 5).

The general trends of the average monthly distribution of dominant pollen types (2017–2020) for both localities are presented in Fig 4. There were no apparent differences in the

**Table 1. Pollen types, annual pollen integrals, maximum (APIn), and percentages recorded in Doha and Al Khor (2017–2020).**

| | DOHA | | | | | | AL KHOR | | | | | | percentage | |
| | 2017–2018 | | 2018–2019 | | 2019–2020 | | 2017–2018 | | 2018–2019 | | 2019–2020 | | 2017–2020 | |
| | APIn | Max | APIn | Max | APIn | Max | APIn | Max | APIn | Max | APIn | Max | Doha | Al Khor |
|---|---|---|---|---|---|---|---|---|---|---|---|---|---|---|
| **Amaranthaceae** | 183 | 8 | 58 | 5 | 585 | 34 | 685 | 36 | 115 | 5 | 923 | 63 | 51.1 | 58.5 |
| **Poaceae** | 39 | 2 | 90 | 4 | 191 | 9 | 52 | 4 | 87 | 5 | 300 | 20 | 19.8 | 15.0 |
| *Prosopis* | 18 | 2 | 14 | 3 | 53 | 9 | 25 | 2 | 9 | 3 | 70 | 14 | 5.3 | 3.5 |
| **Brassicaceae** | 2 | 1 | 14 | 2 | 43 | 12 | 20 | 7 | 20 | 4 | 6 | 2 | 3.6 | 1.6 |
| **Arecaceae** | 13 | 2 | 15 | 1 | 12 | 6 | 19 | 2 | 18 | 2 | 29 | 6 | 2.5 | 2.3 |
| *Plantago* | 11 | 2 | 11 | 3 | 16 | 3 | 14 | 2 | 19 | 2 | 28 | 3 | 2.3 | 2.1 |
| *Artemisia* | 10 | 2 | 1 | 1 | 21 | 5 | 27 | 2 | 7 | 2 | 16 | 5 | 2.0 | 1.7 |
| **Cyperaceae** | 6 | 2 | 14 | 2 | 8 | 2 | 11 | 1 | 27 | 8 | 13 | 2 | 1.7 | 1.7 |
| *Conocarpus* | 5 | 1 | 7 | 1 | 11 | 3 | 28 | 3 | 19 | 3 | 21 | 4 | 1.4 | 2.3 |
| *Casuarina* | 3 | 1 | 11 | 1 | 7 | 2 | 11 | 2 | 3 | 1 | 40 | 6 | 1.3 | 1.8 |
| **Cupressaceae** | 8 | 2 | 4 | 1 | 6 | 2 | 12 | 2 | 3 | 1 | 11 | 4 | 1.1 | 0.9 |
| *Urtica* | 3 | 2 | 4 | 1 | 10 | 3 | 5 | 1 | 6 | 1 | 2 | 2 | 1.1 | 0.4 |
| **Fabaceae** | 0 | 0 | 9 | 2 | 4 | 2 | 0 | 0 | 8 | 1 | 50 | 29 | 0.8 | 2.0 |
| *Olea* | 5 | 1 | 2 | 1 | 3 | 1 | 12 | 3 | 3 | 1 | 5 | 2 | 0.6 | 0.7 |
| **Apiaceae** | 1 | 1 | 0 | 0 | 9 | 2 | 1 | 1 | 0 | 0 | 27 | 5 | 0.6 | 1.0 |
| *Rumex* | 1 | 1 | 3 | 1 | 5 | 2 | 6 | 1 | 3 | 1 | 6 | 2 | 0.6 | 0.5 |
| *Neem* | 2 | 1 | 2 | 1 | 5 | 2 | 0 | 0 | 0 | 0 | 0 | 0 | 0.6 | 0 |
| **Ziziphus** | 6 | 2 | 2 | 1 | 0 | 0 | 5 | 1 | 1 | 1 | 0 | 0 | 0.5 | 0.2 |
| **Compositae** | 2 | 1 | 3 | 1 | 3 | 2 | 2 | 1 | 2 | 1 | 19 | 3 | 0.5 | 0.8 |
| *Echium* | 0 | 0 | 0 | 0 | 6 | 6 | 4 | 1 | 2 | 1 | 4 | 3 | 0.4 | 0 |
| **Myrtaceae** | 1 | 1 | 0 | 0 | 4 | 2 | 1 | 1 | 1 | 1 | 1 | 1 | 0.3 | 0.1 |
| *Ricinus* | 1 | 1 | 3 | 1 | 0 | 0 | 3 | 1 | 4 | 1 | 6 | 2 | 0.2 | 0.4 |
| *Ephedra* | 1 | 1 | 0 | 0 | 2 | 1 | 7 | 2 | 0 | 0 | 11 | 2 | 0.2 | 0.6 |
| *Parkinsonia* | 0 | 0 | 0 | 0 | 0 | 0 | 2 | 1 | 0 | 0 | 0 | 0 | 0.1 | 0 |
| *Arnebia* | 0 | 0 | 0 | 0 | 0 | 0 | 0 | 0 | 2 | 1 | 0 | 0 | 0.8 | 0.0 |
| *Pinus* | 0 | 0 | 0 | 0 | 0 | 0 | 0 | 0 | 0 | 0 | 6 | 2 | 0.2 | 0.0 |
| **Indetermined** | 10 | 6 | 8 | 1 | 7 | 3 | 12 | 2 | 5 | 1 | 9 | 2 | 1.5 | 0.9 |
| **TOTAL** | 332 | - | 275 | - | 1011 | - | 964 | - | 364 | - | 1603 | - | 100 | 100 |

pollen seasons dynamics between the stations of Doha and Al Khor. However, the averages of pollen concentrations for each site gave substantial evidence that higher levels were registered in the Al Khor station, except Brassicaceae and Casuarina pollen which reached higher concentrations in the Doha station.

## Correlation between the total pollen and annual pollen integral

Spearman's correlation tests between the pollen total and the APIn of the dominant types showed a significant positive association between the variables studied in both monitoring stations (P = 0.01), except for differences shown for *Prosopis*, Brassicaceae, *Plantago*, *Artemisia*, Cyperaceae, and *Conocarpus*.

Regarding the Wilcoxon test for comparing non-parametric paired data, the applied test results have shown that no significant differences were detected between the count of selected pollen types either for the pollen types separately or the pollen total (Table 2). Significant differences could be observed for Brassicaceae pollen concentration in both stations (P = 0.000).

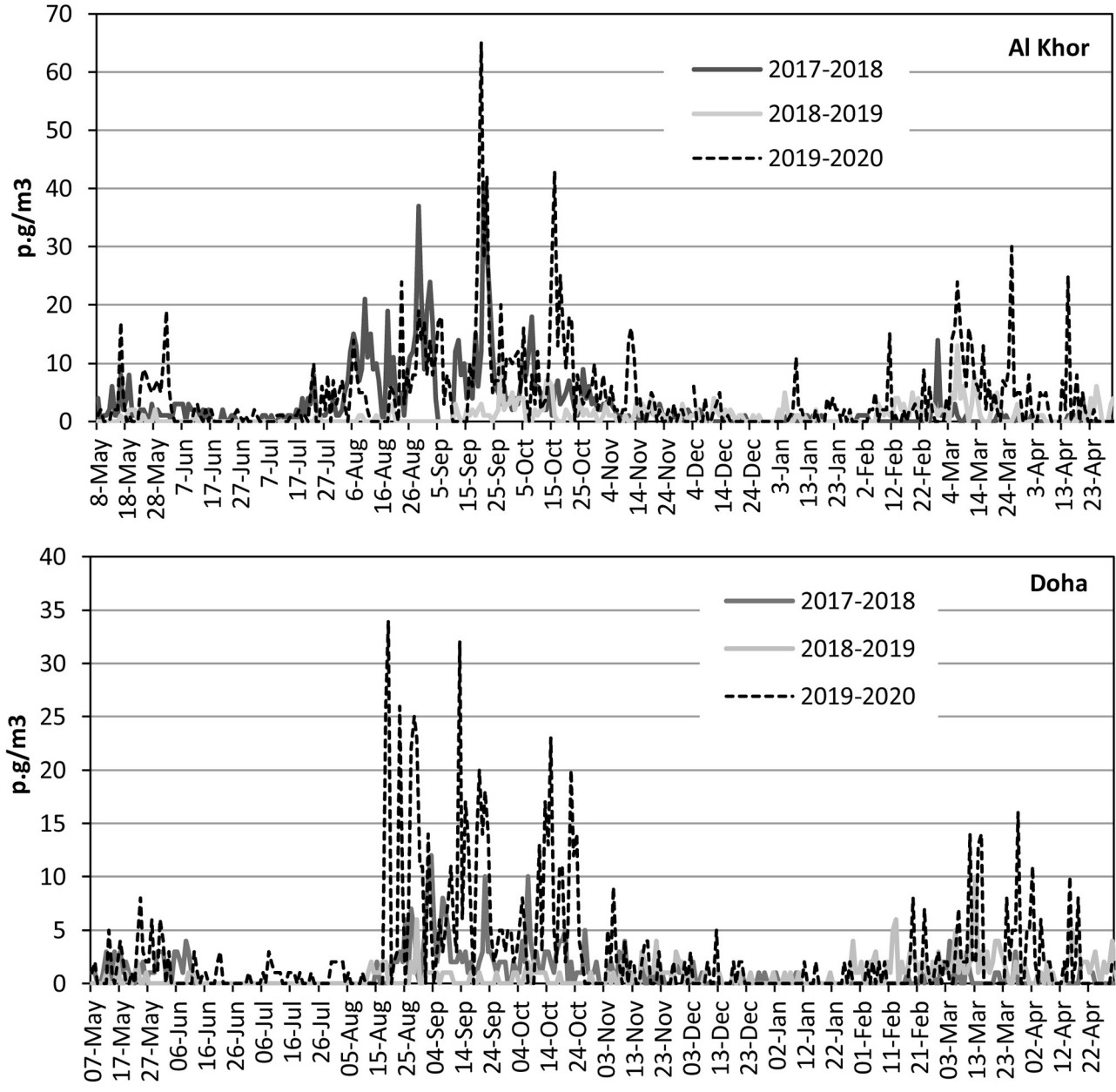

**Fig 3. Seasonal dynamics of total airborne pollen in Doha and Al Khor (2017–2020).**

## Comparison between the two cities pollen *calendars*

Two calendars were created to show the general dynamic followed by the taxa in Al Khor and Doha's atmosphere, which were prepared using the average weekly total pollen count. Only pollen taxa accounting for more than 1% of the annual pollen index were considered.

Significant differences were observed in the number of taxa (Doha 12, Al Khor 11) and the beginning and the end of the respective pollen season. The highest variations in the individual station's pollen levels were observed, with high pollen concentrations (50≥ high) occurring with Amaranthaceae and Poaceae taxa, at the Al Khor station, compared to the Doha station.

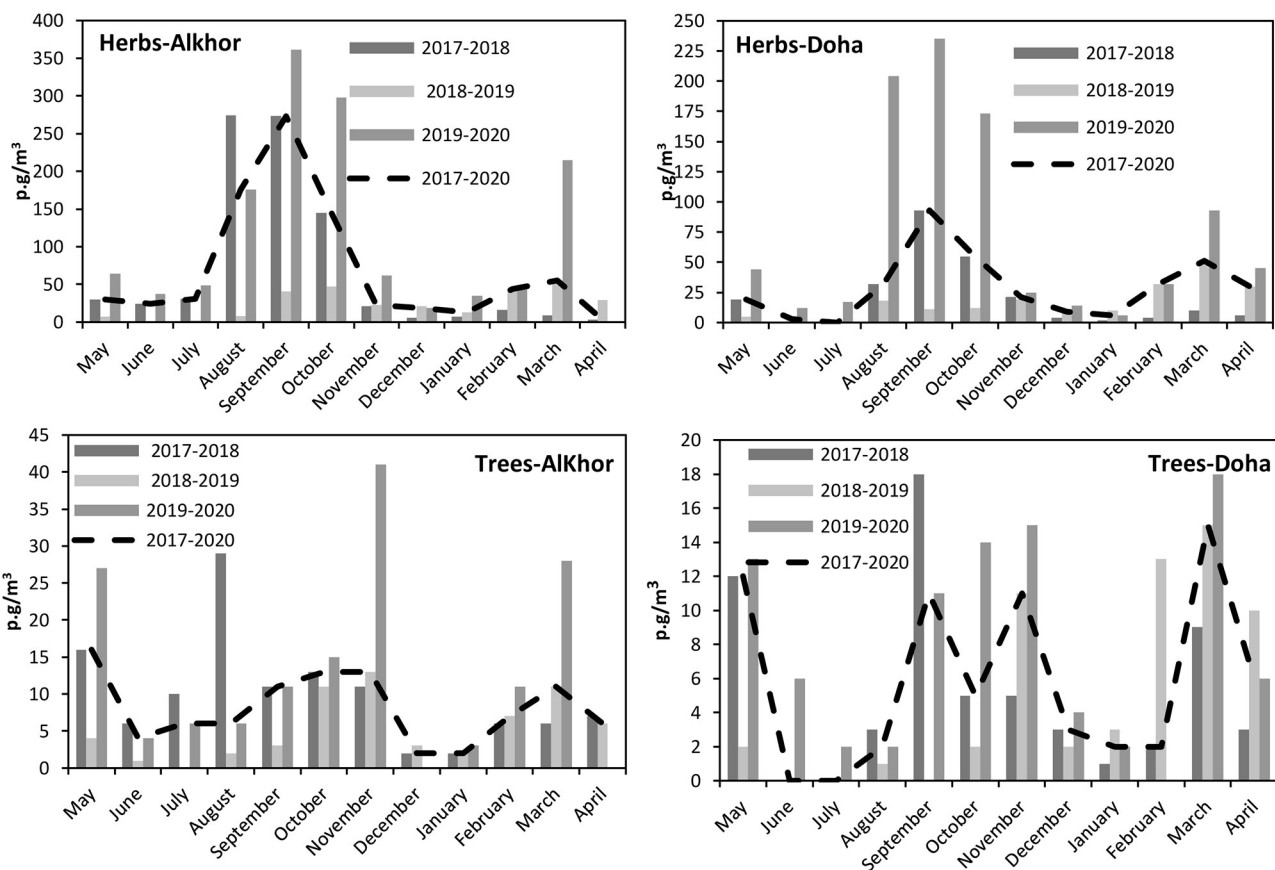

**Fig 4. Monthly variation of trees and herbaceous pollen in the atmosphere of Doha and Al Khor (2017–2020).**

Al Khor's calendar showed that October (8), November (6) were the months of greatest variability regarding the pollen types recorded. Compared to Doha's calendar, a lower abundance of taxa was recorded during October (4) and November (2).

Some pollen types, especially herbaceous species, presented a longer pollen season in the Al Khor station than the Doha station. The pollen grains of Aramanthaceae families were more constant in Al Khor weather, with 39 weeks of pollen season length, longer than the Doha station that showed an average duration in the air as 14 weeks only.

In Al Khor station, moderate levels were observed and reached the first peak in the last week of August and the second peak with the highest level during the third week of September. A different pattern was observed in the Doha station, where pollen grains were abundant with moderate levels from the third week of August until the third week of October.

In comparison, Poaceae pollen was detected in several months, some even being present throughout or, at least, during most of the year. The average weekly abundance at Al Khor station was 28 weeks but was lower at Doha station with 22 weeks (Fig 6).

## Discussion

The three years aerobiological survey in the peninsula of Qatar answered several questions defining the seasonal appearance and measuring the atmospheric concentration of the airborne pollen using two samplers with a distance of around 50 km apart. The increasing awareness of local physicians drove knowledge of the daily, weekly, monthly, and annual pollen

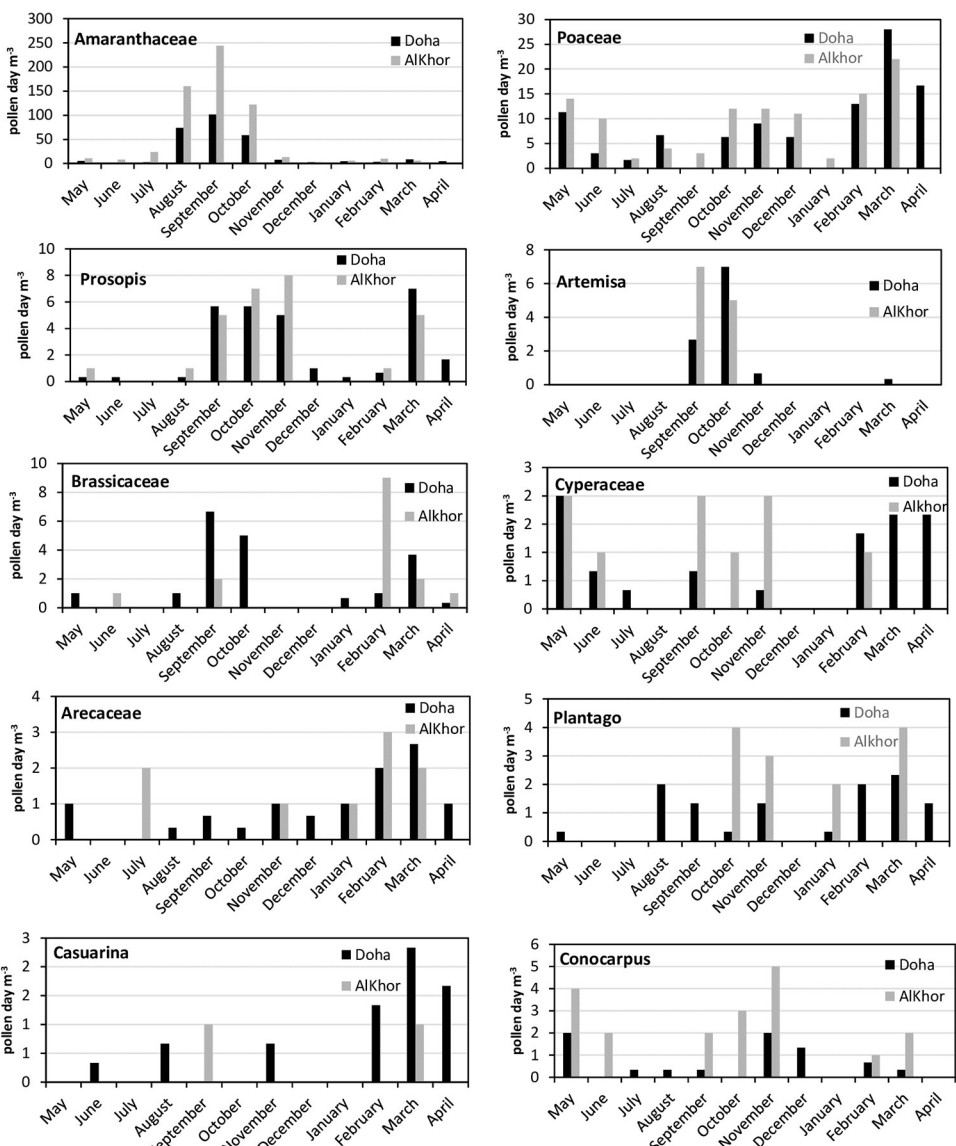

**Fig 5. Average monthly distribution of pollen concentration at Doha and Al Khor stations (2017–2020).**

levels to pollen allergenicity and how that information can be used by patients who have allergies diseases to improve their health and quality of life [30, 31].

A rising number of aerobiological sampling stations are being installed in population centres to collect information on the pollen content of the atmosphere [32] leading to a substantial increase in comparative studies of a different zone within the same city [6, 7] or between different cities [5, 33]. Even though it is usual to find quantitative and qualitative differences in different taxa concentrations, comparative studies have traditionally been made descriptively [34].

The sum of pollen grains in Qatar's atmosphere during the monitored years was significantly higher at Al Khor than at Doha stations. In Al Khor, 25 pollen types (14 herbaceous and 11 trees), while 24 pollen types (14 herbaceous and 10 trees) were recorded in Doha. The frequencies of the significant pollen categories displayed the dominance of herbaceous pollen at

**Table 2. Correlation between the daily mean concentrations obtained for the different pollen types and the total at Doha and Al Khor sampling stations (2017–2020).**

| | Spearman's test | | Wilcoxon's test | |
|---|---|---|---|---|
| | **r** | **P** | **Z** | **P** |
| **Total** | 0.420** | 0.000 | -1.580 | 0.112 |
| **Amaranthaceae** | 0.433** | 0.000 | -1.604 | 0.109 |
| **Poaceae** | 0.272** | 0.000 | -1.069 | 0.280 |
| *Prosopis* | -0,016 | 0.1 | -1.070 | 0.275 |
| **Brassicaceae** | -0.024 | 0.082 | -0.700 | 0.000** |
| **Arecaceae** | 0.181** | 0.000 | -1.604 | 0.109 |
| *Plantago* | 0.079 | 0.07 | -1.604 | 0.109 |
| *Artemisia* | -0.006 | 0.34 | -1.069 | 0.285 |
| **Cyperaceae** | 0.098 | 0.71 | -1.633 | 0.102 |
| *Conocarpus* | -0.036 | 0.02 | -1.604 | 0.109 |
| *Casuarina* | 0.190** | 0.000 | -0.816 | 0.414 |

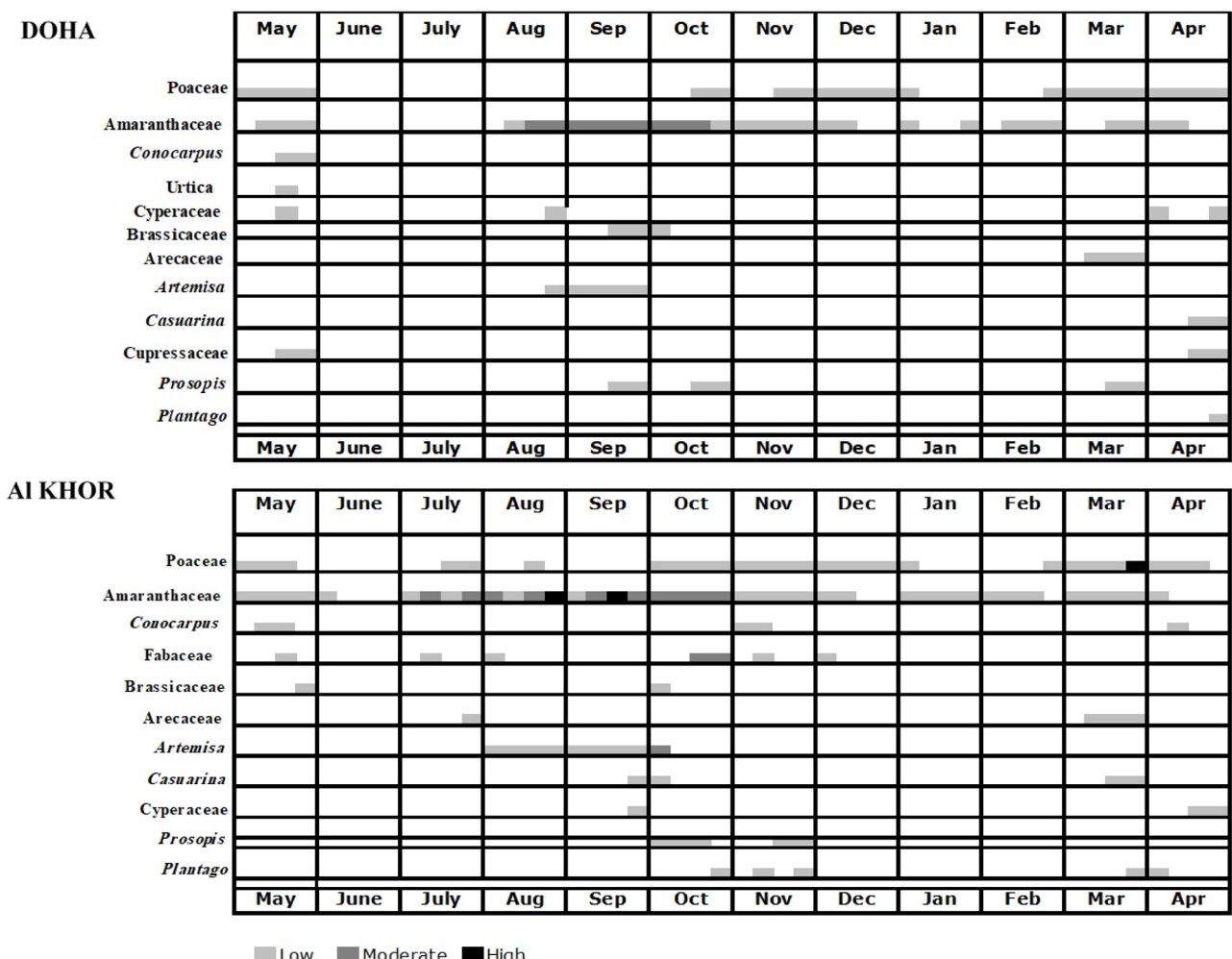

**Fig 6. Pollen calendars of Al Khor and Doha localities (2017–2020).**

Al Khor station with 87%, while trees species remain higher at Doha station with 14% of the total sum (Fig 3). This data provided a direct and synthetic view of the vegetation characters, mainly representing the anemophilous species in and around the sampling stations' geographical locations.

Analysing the pollen index of each pollen type and study year revealed differences in pollen-producing species distribution in the traps' vicinity. In general, the image of the current vegetation obtained from the pollen spectrum analysis is that communities of weeds and grasses are dominant. Herb plants such as Amaranthaceae, Poaceae, and *Plantago* are common in Al Khor compared to Doha. This high representation is due to many species growing as a common perennial weed along the roadsides and natural desert areas in Qatar [23]. Additionally, these findings, which were confirmed using statistical analysis, highlighted the link between airborne pollen counts and predominant local vegetation [34].

The comparison of annual pollen indexes obtained by the two samplers, running at a distance of 50 km, draws particular attention to the pollen concentration's spatial variation. Correlations between the pollen count at the two stations were not statistically significant in most compared taxa such as *Prosopis*, Brassicaceae, *Plantago*, *Artemisia*, Cyperaceae, and *Conocarpus*.

Knowledge of many taxon-specific processes, from plant location and phenology to meteorology and pollen traits, is required to understand the spatiotemporal patterns of pollen observed. Spatiotemporal patterns of pollen concentrations can result from several factors, including spatial variation in plant location and abundance [35]. In a specific way, plant composition within genera can also affect variation in pollen concentrations [36]. Moreover, flowering phenology and meteorology have cause-effect and lead in pollen concentrations influencing pollen release and dispersal at different sites [37, 38].

Knowing the circumstances in which a monitoring station is expected to represent its area would be helpful to researchers, allergy physicians, and the public. Measurement at a single site explained 3–85% of the variation at other sites, depending on the taxon [39]. One approach for reducing exposure errors in allergenic pollen epidemiological studies is to monitor pollen at multiple locations within a city [40]. Monitoring conducted between cites may be a good alternative to the strength of autocorrelation between taxa and allergenic symptoms events.

*Prosopis* was found to be the third predominant pollen type in both stations of Doha and Al Khor. Like many other countries, *Prosopis juliflora* (Sw.), locally known as mesquite, is an invasive plant in the state of Qatar. A native of South and Central America, *Prosopis juliflora* was introduced to many desert environments, including the Gulf countries, mainly to mitigate the effect of desertification [41]. Its allergenicity has been reported in several countries, including the United States [42], India [43], and Saudi Arabia [44]. In this aerobiological study, the relatively low count of *Prosopis* pollen recorded in Al Khor station during the sampling years may be attributed to the number of trees, especially in the trapping site's vicinity. This fact may correspond to date palm or Arecaceae pollen records, which represented 2.5% of the total sum at Doha station and 2.3% at Al Khor station, although higher pollen concentrations were expected to be detected in both traps. Date palm is native and widely grown both in private farms and public areas in Qatar. Qatar ranked the 16th largest date-producing country globally, with more than 580,000 date palm trees growing in an area of 2,469 ha as of 2010. The major cultivation is in the north and central areas of Qatar [45]. The finding of low palm trees pollen concentrations could be due to the distance from the nearest pollen source and pollen sampler since airborne date palm pollen is encountered mainly in the proximity of the trees [46]. Also, it might be due to cultivation practices and artificial pollination of date palm in Qatar and the region [45].

The low representation of other trees in Qatar's pollen spectrum can be explained probably because of their limited distribution and strictly entomophilous nature. Aerobiological studies conducted in the Arabian Gulf indicated the potential allergenicity of locally abundant species and confirmed that both pollen of native and naturalised plant species were represented, of which Amaranthaceae and grasses were reported to be the most common botanical groups triggering respiratory allergic symptoms, followed by *Prosopis*, *Cyperus*, *Plantago*, and Brassicaceae [12, 47].

In our study, two periods of pollen peaks were recorded in Qatar: August-October and March (Fig 1), consistent with Saudi Arabia results [18], but different from Kuwait, where the highest pollen count was recorded April-May and September-October [15].

The data collected in this study showed that the daily mean concentrations of airborne pollen recorded in the studied areas were considered low to moderate. It also implies that allergic diseases due to weed pollen might be most serious in August-September when Amaranthaceae species were abundantly flowering with moderate to highly allergenic levels. The pollen calendars for Doha and Al Khor are typically characteristic of a humid and desert climate, with a limited variety of taxa, only a few of which show a long pollination period, especially of Amaranthaceae and Poaceae, which may last practically a whole year long. In contrast, the pollination period for trees species is short and hardly exceeds three weeks.

The calendars obtained for both localities present similarities with those previously described for Kuwait City [15], Al Khobar, and Abha in Saudi Arabia [18]. The data from pollen calendars of Doha, and Al Khor indicated that potential inter-site differences would therefore appear to be due mainly to differences in the distribution of native and urban flora around the traps.

Several studies reported that a pollen calendar could reveal the relative abundance of certain pollen types over a given period and overcome the expected annual solid variation in number and time of appearance [8, 48]. The current pollen calendar provides baseline data on the prevailing airborne pollen types in two localities in the country. A short series of 3–5 years of continuous air monitoring is recommended permitting a first approach to the aerobiological dynamics of a determinant area[23]. Long-term sampling allows better identification of the periods in which different pollen types present and the determination of any tendencies or variations due to climatic changes [49]. Moreover, seven years is considered the minimum time required to obtain a representative estimation of pollen concentration in any area's atmosphere [50].

From a clinical point of view, pollen grains cause allergic sensitisation and, with continued exposure, lead to symptoms of allergic rhinitis, asthma, and allergic conjunctivitis. The threshold of pollen exposure is considered as the minimum amount of airborne pollen necessary to trigger a nasal or conjunctival allergic reaction may vary widely [51]. Therefore, allergen immunotherapy trials for seasonal pollinosis depend on accurate knowledge of pollen exposure times, estimated by the measure of airborne pollen concentrations (number of pollen in $m^3$ of air) as established by several aerobiologic networks [52]. A threshold of 10 pollen grains/$m^3$ per day is commonly used for allergenicity by the French Network of Aerobiological Monitoring [53] for herbaceous taxa. A reference level of 15–30 pollen grains/$m^3$ per day is the risk threshold established by the Spanish Aerobiology Network [22] to provoke an allergy associated with the presence of these pollen types. In a study from Poland [54], a threshold of grass pollen of 20 pollen grains/$m^3$ was enough to evoke allergic rhinitis symptoms in some patients. In Northeast USA, weed pollen exposures as low as 6–9 pollen grains/$m^3$ were sufficient to trigger asthma symptoms and as low as 2 grains/$m^3$ for grass pollen [55]. In Sweden, nasal symptoms increased linearly with grass pollen count from 0 to 30 grains/$m^3$ [56]. Some other studies have shown that between concentrations of 10 grains/$m^3$ and 14 grains/$m^3$ of grass

pollen exposure, an indication of a threshold value appeared to be more pollen-associated emergency department sits, hospital admissions and drug consumption/medication sales [57].

The current study provided new knowledge on pollen emissions in two localities (Doha and Al-Khor) in the peninsula of Qatar. In our first detailed report on the airborne pollen spectrum in the atmosphere of Qatar (Al-Nesf et al., 2020), results showed that there is a highly significant association between the Amaranthaceae extract SPT results and the symptoms of asthma and allergic rhinitis of patients attending allergic clinics in Qatar and recorded during the pollen season pattern of Amaranthaceae pollen. The main perspective of our current study can be considered a complementary reference to investigate the relationship between the clinical picture of allergic disease and the level of pollen count at which symptoms may be triggered. Consequently, the simplified pollen calendars prepared can assess the allergenic potential of the zone of Doha and Al Khor and be used to those interested in the subject of pollen allergenicity in the peninsula of Qatar.

Some limits in our study must be considered for the aerobiological community. Based on the typical desert ecosystem, dust in the air of Qatar could profoundly hinder pollen trapping due to the inorganic particles of sand that could compete with pollen for the space in the adhesive tape used for pollen collecting. Large quantities of dust particles were observed in the collected tapes. So, pollen concentrations could be underestimated.

## Conclusion

Based on the current three years of aerobiological monitoring, spatial variability of the atmospheric pollen concentration was observed between the sampling stations of Doha and al Khor in the peninsula of Qatar. Inter-site differences were observed for the annual pollen integrals, the recorded peaks, and the main pollen types. Differences were found for certain pollen types, including *Urtica*, Cupressaceae, Fabaceae, confirming the influence of surrounding vegetation on airborne pollen counts at each sampler station. Additional volumetric pollen traps may be needed to characterise the spatiotemporal variation of potential pollen sources in different cities of Qatar and the surrounding region. This reference report can be a fundamental guide for future work in the state of Qatar as well as other countries in the region or countries with a similar atmosphere.

## Author Contributions

**Conceptualization:** Maryam Ali Al-Nesf, Hassan M. Mobayed, Amjad Tuffaha, Mehdi Adeli, Hisham A. Sattar, Maria del Mar Trigo.

**Data curation:** Maryam Ali Al-Nesf, Dorra Gharbi, Ramzy Mohammed Ali, Blessing Reena Dason, Maria del Mar Trigo.

**Formal analysis:** Maryam Ali Al-Nesf, Dorra Gharbi, Ramzy Mohammed Ali, Amjad Tuffaha, Blessing Reena Dason, Mehdi Adeli, Maria del Mar Trigo.

**Funding acquisition:** Maryam Ali Al-Nesf.

**Investigation:** Maryam Ali Al-Nesf, Dorra Gharbi, Hassan M. Mobayed, Ramzy Mohammed Ali, Amjad Tuffaha, Blessing Reena Dason, Hisham A. Sattar, Maria del Mar Trigo.

**Methodology:** Maryam Ali Al-Nesf, Amjad Tuffaha, Hisham A. Sattar, Maria del Mar Trigo.

**Project administration:** Maryam Ali Al-Nesf, Hisham A. Sattar.

**Resources:** Maryam Ali Al-Nesf, Hassan M. Mobayed, Ramzy Mohammed Ali, Amjad Tuffaha, Mehdi Adeli, Maria del Mar Trigo.

**Software:** Maryam Ali Al-Nesf, Ramzy Mohammed Ali, Blessing Reena Dason, Mehdi Adeli, Hisham A. Sattar.

**Supervision:** Maryam Ali Al-Nesf, Hassan M. Mobayed, Amjad Tuffaha, Mehdi Adeli, Hisham A. Sattar, Maria del Mar Trigo.

**Validation:** Maryam Ali Al-Nesf, Dorra Gharbi, Blessing Reena Dason, Mehdi Adeli, Maria del Mar Trigo.

**Visualization:** Maryam Ali Al-Nesf, Dorra Gharbi, Amjad Tuffaha, Blessing Reena Dason, Mehdi Adeli, Maria del Mar Trigo.

**Writing – original draft:** Maryam Ali Al-Nesf, Dorra Gharbi, Ramzy Mohammed Ali, Maria del Mar Trigo.

**Writing – review & editing:** Maryam Ali Al-Nesf, Dorra Gharbi, Hassan M. Mobayed, Ramzy Mohammed Ali, Amjad Tuffaha, Blessing Reena Dason, Mehdi Adeli, Maria del Mar Trigo.

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
