## [Decision Letter · Decision Letter 0]

24 Aug 2021

PONE-D-21-13177

Aerobiological monitoring in a hot geographical region: two sampling stations of two cities (2017-2020)

PLOS ONE

Dear Dr. Al-Nesf,

Thank you for submitting your manuscript to PLOS ONE. After careful consideration, we feel that it has merit but does not fully meet PLOS ONE’s publication criteria as it currently stands. Therefore, we invite you to submit a revised version of the manuscript that addresses the points raised during the review process.

We look forward to receiving your revised manuscript.

Kind regards,

Wei Wu, Ph.D.

Academic Editor

PLOS ONE

Journal Requirements:

2. Thank you for stating the following in the Competing Interests/Financial Disclosure * (delete as necessary) section:

“On behalf of all authors, I disclose that all authors have no competing interests that could be perceived to bias this work.

The authors have declared that no competing interests exist. This manuscript received supported grant funding from the Qatar National Research Fund (QNRF)-Qatar (NPRP 9-241-3-043). However, the funders had no role in study design, data collection and analysis, decision to publish, or preparation of the manuscript.”

We note that you received funding from a commercial source: Qatar National Research Fund

“This paper concluded out of a 3 years grant project. The project “Aerobiological studies in Qatar and Sharjah: Toward the establishment of a network for pollen analysis and allergenicity” (NPRP 9-241-3-043) has been supported by a grant provided by the Qatar National Research Fund (QNRF).”

 “We declare that the research reported in this manuscript received supported grant funding from the Qatar National Research Fund (QNRF)-Qatar (NPRP 9-241-3-043).

MAA received the award grant. URL: https://www.qnrf.org/en-us/Funding/Research-Programs/National-Priorities-Research-Program-NPRP.

Reviewers' comments:

Reviewer's Responses to Questions

**Comments to the Author**

1. Is the manuscript technically sound, and do the data support the conclusions?

Reviewer #1: Partly

Reviewer #2: Yes

2. Has the statistical analysis been performed appropriately and rigorously? 

Reviewer #1: Yes

Reviewer #2: I Don't Know

3. Have the authors made all data underlying the findings in their manuscript fully available?

Reviewer #1: Yes

Reviewer #2: No

4. Is the manuscript presented in an intelligible fashion and written in standard English?

Reviewer #1: Yes

Reviewer #2: Yes

5. Review Comments to the Author

Reviewer #1: The manuscript refers to airborne pollen measurements in two sites in Qatar. To my knowledge, this is the first consistent and continuous biomonitoring ever in that region and for this alone the work deserves scientific merit. I wish to suggest, though, some comments and additions and changes that would further improve the quality of the paper, as follows:

Title:

instead of "a hot geographical region" rephrase into the climatic type the region belongs to, i.e. subtropical desert-type ecosystem, and refer to the country by name.

Abstract:

In some parts it is a bit vague and in some others it is not giving the full information needed. For example:

Background: practically, this is the first measurement in the country for a continuous monitoring interval.

line 5: instead of counts, use measurements or concentrations.

The aim was to assess the abundance and seasonality of the most prevalent pollen types, plus identify potential differences between two sites within the country.

Results, lines 10-12: vague, please omit. What about seasonality (i.e. onset/peak/duration of pollen seasons)?

Conclusions: lines 21-22: this is one, secondary to my opinion, results. Check also my comments above.

Lines 23-26: please omit.

Overall, the abstract needs drastic editing.

line 27: check and replace the "hot" throughout the manuscript.

lines 34-38: i agree. But in lines 35-36, it would be more accurate to be referring to micrometeorological particularities, rather than biogeographical conditions (which refer o much larger spatial scales).

line 41: replace wide with larger.

lines 43-50: the whole concept of pollen calendars seems somewhat over-simplified here. Practically, as mentioned, it is a visualisation of the abundance and seasonality of the most prevalent pollen types. It may indeed be used for symptom prognosis and prophylaxis. But it is not of course directly connected with predictions or with symptoms' relationships, especially in a personalised frame. Please consider editing so that all text becomes more accurate.

Table 1: please check the latin names so that the genera are all in italics.Also, in row 2, delete "(Grass)".

line 161: Urtica in italics.

Overall, the Results are accurate, robust and concise.

lines 327-360: omit completely, unless you show symptom or similar results and associated discussion.

Overall, the Discussion is to-the-point and adequately providing discussion of the current findings, as well as comparisons with previous publications.

Overall comment: even though the use of English language is good, there are still several minor errors throughout the manuscript. Please check once more for details.

Reviewer #2: The research presented in this manuscript appears to have been conducted soundly. The research is significant/important mostly because of the location, Qatar, a country and region where little aerobiological research has been conducted and little is known about the aerobiology or its drivers and associations with allergic respiratory diseases. The manuscript is generally well written, although there are some areas that are in need of attention. The following would improve the manuscript.

It would be useful to include a map of the region showing the location of the two studied cities and some major biogeographic features such as topography, water bodies, major surface types (urban, forest, agricultural etc.).

It would be good to include an extra figure showing the variation in daily temperature and rainfall in the two cities over the study period. While it is understood that the manuscript is not examining relationships between pollen and meteorological factors, at least the inclusion of a figure with the basic temperature and rainfall data would enable a broad interpretation of associations between pollen and these factors to be undertaken.

The title of the manuscript could be more specific by naming the country in which the study is conducted, Qatar, e.g.:

Aerobiological monitoring in a hot geographical region: simultaneous sampling in two

cities in Qatar (2017-2020)

It would be of interest to the aerobiological community to learn of any problems that were encountered in the monitoring associated with the hot temperatures or other factors associated with the location of the study such as occasional high dust loads. For example, did the monitoring equipment or the counting processes encounter any problems associated with these?

Line 100: Please also state at what time of day the pollen trap drums were changed, and in terms of the counting please state how many transects of the slide were counted.

Line 101: Please state the exact date the pollen monitoring was started and finished at each location, i.e., include the day as well as the month and year. Figure 1 suggests the sampling started on 8 and 7 May 2017 in Al Khor and Doha, respectively. Is this the case?

Line 114: Change “undermined” to “undetermined”.

Table 1: The table caption should also state that maximums are included in the table. Also in relation to the maximums, a column total is included for each maximum column which doesn’t really make sense, so I suggest not including a column total for the maximum columns.

Table 2: The table caption needs to be more specific, such as stating the names of the two locations, the time period over which the correlations are being done, and if it is daily data that are being correlated. In the Spearman’s test P column, delete the “NS”. This is not required, given the statistically significant level was defined in the Methods section.

Figure 4: The bands indicating low, moderate, and high currently only fill part of the height of each cell. This makes it difficult to clearly see when each type of pollen is present. I suggest the full depth of each cell be shaded if the pollen is present. Also, the font size for the different taxa could be increased so that they are easier to read.

Lines 337-352: Two studies have been published recently that are directly relevant to the discussion in this paragraph and should therefore be included in that discussion:

Steckling-Muschack, N., Mertes, H., Mittermeier, I. et al. A systematic review of threshold values of pollen concentrations for symptoms of allergy. Aerobiologia 37, 395–424 (2021). https://doi.org/10.1007/s10453-021-09709-4

Becker, J., Steckling-Muschack, N., Mittermeier, I. et al. Threshold values of grass pollen (Poaceae) concentrations and increase in emergency department visits, hospital admissions, drug consumption and allergic symptoms in patients with allergic rhinitis: a systematic review. Aerobiologia (2021). https://doi.org/10.1007/s10453-021-09720-9

6. PLOS authors have the option to publish the peer review history of their article (what does this mean?). If published, this will include your full peer review and any attached files.

Reviewer #1: No

Reviewer #2: No

---

## [Author Response · Author response to Decision Letter 0]

10 Apr 2022

Response to Review Comments to the Author

Reviewer #1: The manuscript refers to airborne pollen measurements in two sites in Qatar. To my knowledge, this is the first consistent and continuous biomonitoring ever in that region and for this alone the work deserves scientific merit. I wish to suggest, though, some comments and additions and changes that would further improve the quality of the paper, as follows:

Title:

instead of "a hot geographical region" rephrase into the climatic type the region belongs to, i.e. subtropical desert-type ecosystem, and refer to the country by name.

Thank you for your valuable comment. A modification was made to the title adding the word desert-type ecosystem and highlighting the study country name.

Abstract:

In some parts it is a bit vague and in some others it is not giving the full information needed. For example:

*Background: practically, this is the first measurement in the country for a continuous monitoring interval.

Thank you for your comment. Information was modified as suggested.

*line 5: instead of counts, use measurements or concentrations.

Thank you for your comment. The word “counts” was modified to concentrations.

*The aim was to assess the abundance and seasonality of the most prevalent pollen types, plus identify potential differences between two sites within the country.

Thank you for the suggestion. The suggested sentence was added to the abstract.

*Results, lines 10-12: vague, please omit. What about seasonality (i.e. onset/peak/duration of pollen seasons)?

Thank you for this valuable comment. The detailed information was added to the abstract in reference to the seasonality and duration of pollen seasons observed between the sampling stations.

*Conclusions: lines 21-22: this is one, secondary to my opinion, results. Check also my comments above.

Thank you for your review. The conclusion section was reviewed.

*Lines 23-26: please omit.

Overall, the abstract needs drastic editing.

Thank you for your comment. The abstract was edited and reviewed.

*line 27: check and replace the "hot" throughout the manuscript.

Thank you for your comment. The word was modified to desert-type ecosystem.

*lines 34-38: i agree. But in lines 35-36, it would be more accurate to be referring to micrometeorological particularities, rather than biogeographical conditions (which refer o much larger spatial scales).

Thank you for your valuable comment. We agree with you. Our study focused on a specific subtropical desert-type ecosystem, with particular meteorological conditions limited to a small scale of the area.

*line 41: replace wide with larger. 

 Thank you for your comment. The word is replaced.

*lines 43-50: the whole concept of pollen calendars seems somewhat over-simplified here. Practically, as mentioned, it is a visualisation of the abundance and seasonality of the most prevalent pollen types. It may indeed be used for symptom prognosis and prophylaxis. But it is not of course directly connected with predictions or with symptoms' relationships, especially in a personalised frame. Please consider editing so that all text becomes more accurate.

Thank you for your valuable comment. We understand that the concept of pollen calendars is somewhat over-simplified here. First of all, our study is the first allergenic pollen monitoring in Qatar and results of three years. The paragraph in the introduction section about the pollen calendar concept was edited and updated. 

*Table 1: please check the latin names so that the genera are all in italics.Also, in row 2, delete "(Grass)". 

Thank you for your comment. The word Grass was deleted, and all the genera were changed to italics.

*line 161: Urtica in italics. 

Thank you for your comment. The format was corrected.

Overall, the Results are accurate, robust, and concise.

lines 327-360: omit completely, unless you show symptom or similar results and associated discussion.

Thank you for your valuable comment. Our study is basically a pioneer project in Qatar. The aerobiological data are limited in the Arab peninsula, sharing common micrometeorological indicators. In this context, our discussion was associated with similar results from other aerobiological surveys. In our case, a previous study (Al-Nesf et al., 2020) showed respiratory allergic symptoms, and the correlation of these symptoms with allergenic airborne pollen in the atmosphere of Qatar is added to the manuscript to confirm the association between the frequently airborne pollen (Amaranthaceae) detected by aerobiological monitoring samplers in respiratory allergy symptoms (asthma and allergic rhinitis).

Overall, the discussion is to-the-point and adequately providing discussion of the current findings, as well as comparisons with previous publications.

Thank you for your comment.

Overall comment: even though the use of English language is good, there are still several minor errors throughout the manuscript. Please check once more for details.

Thank you for your comment. The whole manuscript was revised.

Reviewer #2: The research presented in this manuscript appears to have been conducted soundly. The research is significant/important mostly because of the location, Qatar, a country, and region where little aerobiological research has been conducted and little is known about the aerobiology or its drivers and associations with allergic respiratory diseases. The manuscript is generally well written, although there are some areas that are in need of attention. The following would improve the manuscript.

It would be useful to include a map of the region showing the location of the two studied cities and some major biogeographic features such as topography, water bodies, major surface types (urban, forest, agricultural etc.).

Thank you for your valuable comments. A topographic map was added to the figures list (Figure 1). A map of Qatar showing the main sampling locations. The common type of landscape of Qatar peninsula is rocky desert, depressions, and salt marshes, and in general, it is flat to undulating. The highest point is called Aba-el-Baul (105 m). There are no rivers, creeks or lakes in Qatar. However, some groundwaters come to the surface as springs in oases

*It would be good to include an extra figure showing the variation in daily temperature and rainfall in the two cities over the study period. While it is understood that the manuscript is not examining relationships between pollen and meteorological factors, at least the inclusion of a figure with the basic temperature and rainfall data would enable a broad interpretation of associations between pollen and these factors to be undertaken.

Thank you for your comment. An extra figure of meteorological spatial patterns of the sampling locations was added to the manuscript. (Figure2)

*The title of the manuscript could be more specific by naming the country in which the study is conducted, Qatar, e.g.:

Aerobiological monitoring in a hot geographical region: simultaneous sampling in two cities in Qatar (2017-2020).

Thank you for your comment. The title was modified as suggested.

*It would be of interest to the aerobiological community to learn of any problems that were encountered in the monitoring associated with the hot temperatures or other factors associated with the location of the study such as occasional high dust loads. For example, did the monitoring equipment or the counting processes encounter any problems associated with these?

Thank you for your comments. In fact, the aerobiological sampling in Qatar, a typical country with a desert ecosystem, was associated with the problem of dust loads. Large quantities of dust particles were observed in the collected tapes during several days of the year. This issue can be one of the major problems in aerobiological air monitoring in similar areas.

*Line 100: Please also state at what time of day the pollen trap drums were changed, and in terms of the counting please state how many transects of the slide were counted.

Thank you for your comments. The requested information was added to the manuscript.

*Line 101: Please state the exact date the pollen monitoring was started and finished at each location, i.e., include the day as well as the month and year. Figure 1 suggests the sampling started on 8 and 7 May 2017 in Al Khor and Doha, respectively. Is this the case?

Thank you for your comment. The sampling was started in Doha station on 7th of May 2017 and in Al Khor station on the 8th of May 2017. This information was added to the manuscript.

*Line 114: Change “undermined” to “undetermined”.

Thank you for your comment. The word was changed. 

*Table 1: The table caption should also state that maximums are included in the table. Also in relation to the maximums, a column total is included for each maximum column which doesn’t really make sense, so I suggest not including a column total for the maximum columns.

Thank you for your comments. The table was revised. 

*Table 2: The table caption needs to be more specific, such as stating the names of the two locations, the time period over which the correlations are being done, and if it is daily data that are being correlated. In the Spearman’s test P column, delete the “NS”. This is not required, given the statistically significant level was defined in the Methods section.

Thank you for your comment. The table was revised.

*Figure 4: The bands indicating low, moderate, and high currently only fill part of the height of each cell. This makes it difficult to clearly see when each type of pollen is present. I suggest the full depth of each cell be shaded if the pollen is present. Also, the font size for the different taxa could be increased so that they are easier to read.

*Lines 337-352: Two studies have been published recently that are directly relevant to the discussion in this paragraph and should therefore be included in that discussion:

Steckling-Muschack, N., Mertes, H., Mittermeier, I. et al. A systematic review of threshold values of pollen concentrations for symptoms of allergy. Aerobiologia 37, 395–424 (2021). https://doi.org/10.1007/s10453-021-09709-4

Becker, J., Steckling-Muschack, N., Mittermeier, I. et al. Threshold values of grass pollen (Poaceae) concentrations and increase in emergency department visits, hospital admissions, drug consumption and allergic symptoms in patients with allergic rhinitis: a systematic review. Aerobiologia (2021). https://doi.org/10.1007/s10453-021-09720-9

Thank you for your suggestions. References are included in the manuscript.

---

## [Decision Letter · Decision Letter 1]

22 Jun 2022

Aerobiological monitoring in a desert-type ecosystem : two sampling stations of two cities (2017-2020) in Qatar

PONE-D-21-13177R1

Dear Dr. Al-Nesf,

We’re pleased to inform you that your manuscript has been judged scientifically suitable for publication and will be formally accepted for publication once it meets all outstanding technical requirements. We also suggest you address the minor comments from the two reviewers during the publication process.

Kind regards,

Wei Wu, Ph.D.

Academic Editor

PLOS ONE

Additional Editor Comments (optional):

Reviewers' comments:

Reviewer's Responses to Questions

**Comments to the Author**

1. If the authors have adequately addressed your comments raised in a previous round of review and you feel that this manuscript is now acceptable for publication, you may indicate that here to bypass the “Comments to the Author” section, enter your conflict of interest statement in the “Confidential to Editor” section, and submit your "Accept" recommendation.

Reviewer #1: All comments have been addressed

Reviewer #2: All comments have been addressed

2. Is the manuscript technically sound, and do the data support the conclusions?

Reviewer #1: Yes

Reviewer #2: Yes

3. Has the statistical analysis been performed appropriately and rigorously? 

Reviewer #1: Yes

Reviewer #2: Yes

4. Have the authors made all data underlying the findings in their manuscript fully available?

Reviewer #1: Yes

Reviewer #2: Yes

5. Is the manuscript presented in an intelligible fashion and written in standard English?

Reviewer #1: Yes

Reviewer #2: Yes

6. Review Comments to the Author

Reviewer #1: The authors have efficiently addressed all reviewers' concerns and recommendations and, having done so, the quality of the manuscript has greatly improved.

There are still some minor linguistic mistakes, which the authors need to fix during the proof-reading process, after they carefully go through the manuscript once more (lines 111-112, 119, 199, etc).

Table 2: mark the significant relationships, otherwise omit the non-significant.

Reviewer #2: The authors appear to have made a good attempt at addressing the comments of both reviewers. The only further comments I have are the following three minor corrections which I expect could be corrected by the journal, without further need for the authors to further revise the manuscript.

In the article title, the phrase “in Qatar” should appear before the years, not after. That is, change:

… sampling stations of two cities (2017-2020) in Qatar

to:

… sampling stations of two cities in Qatar (2017-2020)

In the Table 1 heading, the abbreviation for annual pollen integral needs to be moved from after the word “maximum” to after the term “annual pollen integrals”. That is, change:

Table 1. Pollen types, annual pollen integrals, maximum (APIn), and percentages …

to:

Table 1. Pollen types, annual pollen integrals (APIn), maximum, and percentages …

In the Abstract, the word “Combined,” needs to be added at the beginning of the following sentence:

“Amaranthaceae and Poaceae airborne pollen constituted 73.5% and 70.9% of the total amount of pollen detected at the samplers of Al Khor station and Doha station.”

7. PLOS authors have the option to publish the peer review history of their article (what does this mean?). If published, this will include your full peer review and any attached files.

Reviewer #1: **Yes: **Athanasios Damialis

Reviewer #2: No

---

## [Editor Report · Acceptance letter]

27 Jun 2022

PONE-D-21-13177R1 

Aerobiological monitoring in a desert type ecosystem: two sampling stations of two cities (2017-2020) in Qatar 

Dear Dr. Al-Nesf:

I'm pleased to inform you that your manuscript has been deemed suitable for publication in PLOS ONE. Congratulations! Your manuscript is now with our production department. 

Kind regards, 

on behalf of

Dr. Wei Wu 

Academic Editor

PLOS ONE